# GRDC-Caravan: extending Caravan with data from the Global Runoff Data Centre

Claudia Färber[1], Henning Plessow[1], Simon Mischel[1], Frederik Kratzert[2], Nans Addor[3,4], Guy Shalev[5], Ulrich Looser[1]

[1] Global Runoff Data Centre (GRDC), Federal Institute of Hydrology (BfG), Koblenz, 56068, Germany
[2] Google Research, Vienna, 1010, Austria
[3] Fathom, Bristol, BS8 1EJ, UK
[4] University of Exeter, Exeter, EX4 4RJ, UK
[5] Google Research, Tel Aviv, 6789141, Israel

*Correspondence to*: Claudia Färber (faerber@bafg.de)

**Abstract.** Large-sample datasets are essential in hydrological science to support modelling studies and advance process understanding. Here, we present the GRDC-Caravan dataset, an extension to the large-sample hydrology project Caravan. Caravan is a community initiative, which aims to combine large-sample hydrology datasets of meteorological forcing data, catchment attributes, and discharge data for catchments around the world. The GRDC-Caravan extension is based on a subset of hydrological discharge data and station-based watersheds from the Global Runoff Data Centre (GRDC), which are covered by an open data policy. The GRDC is an international data centre operating under the auspices of the World Meteorological Organization (WMO), which collects quality-controlled river discharge data and associated metadata from the National Meteorological and Hydrological Services (NMHS) of WMO Member States. The extension contains discharge data and catchment boundaries from GRDC, which can be released under a permissive license (CC-BY-4.0). In addition, the extension contains meteorological forcing data and catchment attributes from the global datasets ERA5-Land and HydroATLAS in a standardized format. The dataset covers stations from 5,356 catchments and 25 countries and spans the years 1950 – 2023. Compared to the core version of Caravan, the extension takes the total number of Caravan catchments to 22,372 (of which 1589 catchments are duplicates between the core and extensions). While in the core Caravan dataset mostly stations from Northern America, Central Europe and South America were included, the new extension significantly improves the global coverage of the dataset with new stations across Europe, South America, South Africa, Australia and New Zealand. In addition, the temporal extension of the time series could be significantly increased from 40 to 70 years. The extension strongly improves the global and temporal coverage of Caravan and represents a valuable dataset for global hydrological and climatological modelling studies. The dataset is released under a CC-BY-4.0 license that allows redistribution and is publicly available on Zenodo: https://doi.org/10.5281/zenodo.15349031 (Färber et al., 2025).

# 1 Introduction

River systems are an integral part of the global water cycle, which are linked to many processes on local, regional and global scales (Dorigo et al., 2021). Observational river discharge data are counted as Essential Climate and Water Variables being fundamental for a wide range of applications such as flood and drought management, the modelling of the global water balance, the analysis of long-term circulation patterns, and the estimation of fluxes into the oceans (Lawford et al., 2023; GCOS, 2021; GEO, 2014). The largest hydrological database for observational in situ discharge data is the Global Runoff Database, operated by the Global Runoff Data Centre (GRDC). The GRDC is an international data centre, which operates under the auspices of the World Meteorological Organization (WMO) at the German Federal Institute of Hydrology (BfG). GRDC provides daily and monthly time series data from the National Meteorological and Hydrological Services of WMO member states, which can be downloaded for non-commercial use, but the data remains property of the owner and redistribution is not allowed (GRDC, 2025a).

In parallel to hydrological data made available by GRDC and water services (e.g., US Geological Survey, USGS), river discharge data is increasingly being provided through large-sample hydrology (LSH) datasets collated by third parties. There are global collections of streamflow data such as the Global Streamflow Indices and Metadata Archive (GSIM), which provides global streamflow indices and metadata (Do et al., 2018; Gudmundsson et al., 2018). Other datasets, such as MOPEX (Schaake et al., 2006), CAMELS (Newman et al., 2015; Addor et al., 2017) and HYSETS (Arsenault et al., 2020) are based on flow time series, which are complemented by atmospheric forcing (e.g., precipitation) time series and catchment attributes (characterizing e.g., the dominant soil type or the level of anthropogenic water extractions). These time series and attributes opened new possibilities in hydrology, enabling for instance the development of machine learning based hydrological models (e.g. Kratzert et al., 2019a; Kratzert et al., 2019b) and the identification of human impacts on river flows over large regions (e.g. Bloomfield et al., 2021; Chagas et al., 2022).

While these datasets have led to key advances, progress is hindered by a few shortcomings (Addor et al., 2020). One of them is the lack of common standards, which makes their combination challenging. The Caravan project (Kratzert et al., 2023) was designed to overcome this and to create a global dataset by combining seven existing LSH datasets (Newman et al., 2015; Fowler et al., 2021; Chagas et al., 2020; Alvarez-Garreton et al., 2018; Coxon et al., 2020; Arsenault et al., 2020; Klingler et al., 2021). We refer to this collection of catchments as the "core Caravan dataset" below. In addition to this dataset, Caravan is also a cloud-based workflow (Fig. 1) which allows members of the hydrology community to extend the dataset to new locations, generating what we refer to as "extensions". Meteorological forcing data and catchment attributes for all Caravan catchments (the core dataset and extensions) are all extracted from the global datasets ERA5-Land (Muñoz-Sabater et al., 2021) and HydroATLAS (Lehner et al., 2019; Linke et al., 2019), guaranteeing spatial consistency and comparability across regions. They also use the exact same data format, which facilitates their uptake by the community and their combination.

Currently, the Global Runoff Database contains river discharge data from more than 10,000 stations in 160 countries, dating back up to 200 years. The core Caravan dataset has a strong focus on North America, South America, Europe and Australia,

although several extensions have been produced since its release covering different spatial and hydroclimatic regions of the world (Koch, 2022; Morin, 2024; Höge et al., 2023; Casado Rodríguez, 2023; Helgason and Nijssen, 2024, Dolich et al. 2024). The aim of this study was to extend the spatial and temporal coverage of Caravan with GRDC discharge data, which can be released with a permissive license (CC-BY-4.0). The extension was created using the cloud-based workflow of the Caravan framework and provides the first subset of freely available GRDC discharge data together with meteorological forcing data and catchment attributes.

## 2 Methodology

The GRDC-Caravan extension was developed following the Caravan methodology in Kratzert et al. (2023). A simplified workflow, how this and other extensions were built is shown in Fig. 1. Meteorological forcing data, catchment attributes and river discharge were compiled using Google Earth Engine in the cloud (Kratzert 2025). A detailed description of the dataset structure and the processing of the dataset components is provided below.

### 2.1 Dataset structure

The current version of the GRDC-Caravan extension dataset is available at https://doi.org/10.5281/zenodo.15349031. Due to its size, the dataset is provided in two zip archives:

1. GRDC_Caravan_csv.zip: provides the time series data as comma-separated text files (CSV) (downloadable as 8.8 GB zip archive)
2. GRDC_Caravan_nc.zip: provides the time series data in the Network Common Data Form (NetCDF) (downloadable as 7.6 GB zip archive)

The data in both versions are identical, but users can choose if they require the time series data in CSV or NetCDF format. The organization of the subfolders corresponds to the core Caravan dataset of Kratzert et al. (2023):

- The attributes folder contains a subfolder with four csv (comma-separated values) files. The file 'attributes_hydroatlas_grdc.csv' contains attributes derived from HydroATLAS and the file 'attributes_caravan_grdc.csv' contains climate indices derived from ERA5- Land. The metadata information is listed in 'attributes_other_grdc.csv', while the last file 'attributes_additional_grdc.csv' is specific for the GRDC extension. It contains attributes from GRDC (Table 1) including the original national station IDs ('nat_id') and catchment information. The first column in all attributes file is called 'gauge_id' and contains a unique station identifier of the source dataset (GRDC) and the station id as defined in the original source dataset. The attribute 'nat_id', which is the national station code for each gauge station, allows the user to find duplicates between GRDC station codes and stations that are already included in Caravan.

- The shapefiles folder contains a subfolder with a shapefile with the catchment boundaries of each station within the dataset. This shapefile was used to derive the catchment attributes and ERA5-Land time series data. Each polygon in a given shapefile has a field 'gauge_id' that contains the unique station identifier.

- The timeseries folder contains one subfolder (CSV or NetCDF, depending on the chosen file format). Within the subdirectory, there is one file (either CSV or NetCDF) per basin, containing all time series data (meteorological forcings, state variables, and streamflow). The netCDF files also contain metadata information, including physical units, timezones, and information on the data sources.

- The licenses folder contains license information of all data included in the extension.

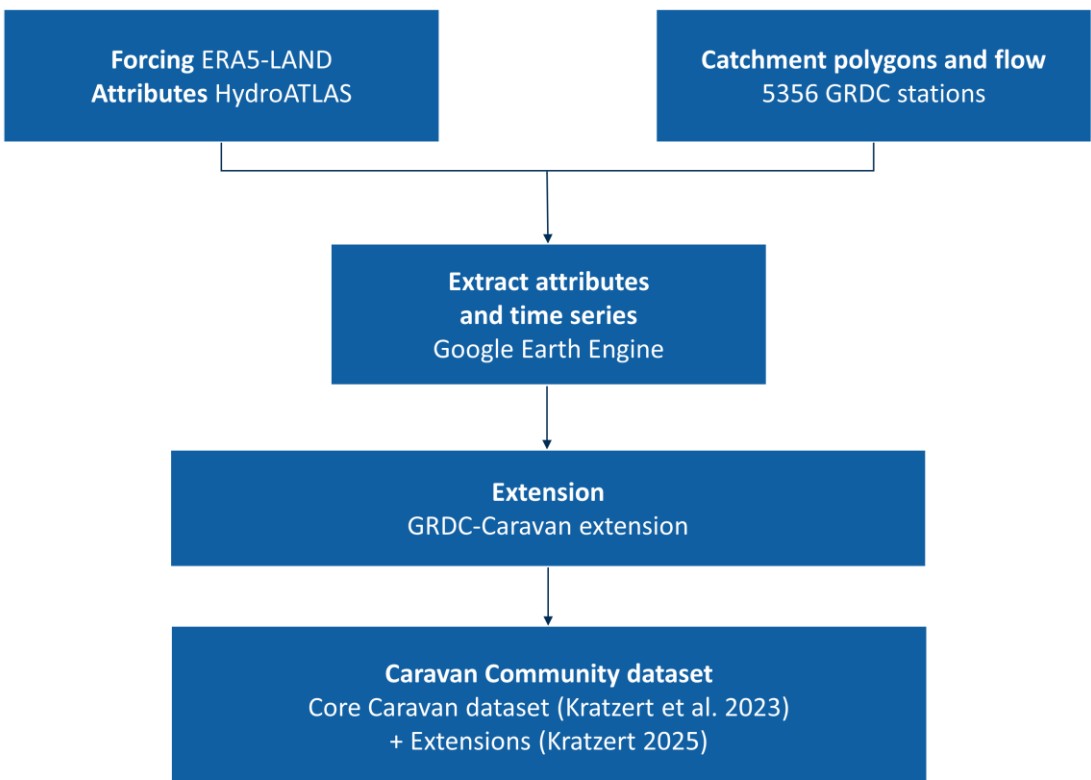

**Figure 1: Flow-chart demonstrating the generation of Caravan datasets as described in Kratzert et al. (2023), using this extension as an example. All Caravan datasets are composed of forcing data, attributes, catchment polygons and flow data. The code to create new extensions is available online and all data is processed in the cloud. See Kratzert (2025) for published Caravan extensions.**

## 2.1 River discharge data

For the compilation of the dataset, river discharge time series from the Global Runoff Data Centre (GRDC) were used. Most time series from GRDC are provided by the NMHS of WMO member states. In exceptional cases, the data is also provided by research institutions (e.g. Greenland). Although there are now several APIs available that allow an automated import of time series, the GRDC usually receives updated time series at irregular intervals and often only upon request.

Basically, there are two different types of updates. In the first case, we receive time series that are consecutive of the previous one, which only are appended. In the second case, we receive historical time series data, which are then compared to the old one in our database. The quality of the time series data is then inspected both numerically and graphically.

During the validation, the provided streamflow values are compared individually for each day. If the new value deviates from the previous one by more than 10 %, the old and new time series are graphically generated, overlaid, and reviewed. This only happens occasionally, but in most cases, the rating curve has been recalculated, resulting in differences in streamflow peaks on the same day. If a deviation cannot be explained or is incorrect, the GRDC consults with the NMHS. In rare cases, this may lead to a revised and improved time series being delivered.

As Caravan is a truly open source initiative (Kratzert et al., 2023), only those stations whose countries adhere to an open data policy with a permissive data sharing license were included in the dataset. In addition, only stations supported by a polygon of the catchment area were chosen. Furthermore, the stations must contain daily discharge values with a time series of at least 1 year and should end in 2010 or more recently. As for the rest of Caravan, discharge data is provided in area normalized units (mm/day). We used the area of the catchment polygons for the normalization, which is provided as the area in the catchment attribute.

**Table 1: Additional attributes that are included in the GRDC-Caravan extension in addition to the set of attributes in the core Caravan dataset (Kratzert et al. 2023).**

| Attribute | Description | Unit |
|-----------|-------------|------|
| gauge_id | Unique station identifier | - |
| nat_id | National station id | - |
| wmo_reg | WMO region | - |
| sub_reg | WMO subregion (basin) | - |
| country | Country code (ISO 3166) | - |
| area_shp | Catchment size, as derived catchment polygons | km$^2$ |
| altitude | Height of gauge zero | m.asl |
| lat_pp | Latitude of pour point, decimal degree | - |
| long_pp | Longitude of pour point, decimal degree | - |
| dist_km | Distance between original and new position | km |

| quality | Catchment Quality indicator | - |
|---------|------------------------------|---|
| type | Catchment type: Automatic or Manual | - |
| comment | Catchment Comment | - |
| source | Source: MERIT or HydroSHEDS | - |
| d_start | Daily data available since | year |
| d_end | Daily data available until | year |
| d_yrs | Length of time series of daily data | - |
| d_miss | Percentage of missing values | - |
| lta_discharge | Long-term average discharge | $m^3s^{-1}$ |
| r_vol_yr | Mean annual volume | $m^3s^{-1}$ |
| r_height_yr | Mean annual runoff depth | mm |

## 2.2 Catchment polygons and gauge locations

130 All stations that meet the criteria for inclusion in the GRDC-Caravan extension, contain a polygon of the station's catchment area. The polygons are mandatory, as they will later be used to derive meteorological forcing data (see chapter 2.3). Below a detailed description of the calculation and evaluation of the catchment areas and the station allocation is given. Additionally, a quality indicator for the calculated catchments is provided.

### 2.2.1 Area calculation

135 For the calculation of station catchment areas, a global flow direction raster was chosen, preferably with an associated river network. The initial dataset utilized is the HydroSHEDS dataset with a 15 arc-second resolution, which corresponds to approximately 500 m at the equator. The dataset has almost global coverage, but was supplemented with the Hydro1k data set above 60° northern latitude, where the quality for these catchment areas drops considerably (Lehner et al., 2008). Therefore, the Multi-Error-Removed Improved-Terrain (MERIT) Hydro dataset was used additionally. MERIT has a 3 arc-second

140 resolution, equivalent to about 90 meters at the equator (Yamazaki et al., 2019). This dataset is well-suited for higher latitudes and particularly for smaller catchment areas. Lin et al. (2019) employed the TauDEM tool to derive the river network of MERIT Hydro.

For the calculation of the upstream area of the stations with the HydroSHEDS dataset the self-developed R package 'GRDCFlowTools' has been used (GRDC, 2025b). It contains functions to process raster files and to determine all upstream

145 cells of a given point, using among others the R package igraph (Csardi and Nepusz, 2006), which is a library for graph theory and network analysis based on the principle of topological sort. Creating these graph objects requires a lot of computing

memory and quickly reaches its limit, when using raster sets covering a whole continent. The function createSplitFlowGraph of the self-developed R package was used to divide a flow direction raster into any number of stripes including their graph objects. For these calculations a separation of the HydroSHEDS flow direction raster into 10 parts seemed to be a reasonable amount. With the created graph-objects the function getBasinFromSplitGraph derives all inflow cells to a given cell number of a flow direction raster from which the total area was determined. This procedure was done for each pixel on the river network.

For the area calculations with the MERIT Hydro dataset the delineator.py Python scripts of Heberger (2021) were used, which have already proven effective in other LSH datasets (Loritz et al., 2024; Senent-Aparicio et al. 2024). It allows quick watershed delineation, using a hybrid of vector- and raster-based methods. It consists of two modes: A "high-resolution" mode, based on MERIT Hydro and a "low-resolution" mode based on HydroSHEDS. We used the "high-resolution" mode, which requires only a column-based file with a unique id and the coordinates to run the script. The huge advantage of the method is how fast the raster is read and processed. This is due to the python package 'pysheds', which is able to read the raster files using a bounding box, where only a particular part of the raster grid is processed. The bounding box is thereby created by the intersection of the unit catchment and the outlet point, clipping the flow direction and accumulation grids to the extent of the unit catchment. To achieve this, all cells outside the unit catchment are marked as 0 to prevent that a neighbouring catchment is considered. Before the catchment is calculated with the grid.catchment function of 'pysheds' the point is snapped to the nearest stream, which depends on the number of upstream pixels to define a waterway. For smaller catchments a smaller number is recommended and vice versa for large ones.

## 2.2.2 Catchment evaluation and station allocation

The methodology for calculating and evaluating the station catchment areas was adopted from the technical report of Lehner (2012). To link the gauging stations to the river network, a two-fold strategy was used. First an automatic station allocation was carried out, where a 5 km buffer around each station has been drawn. Within this buffer, points for each cell of the flow direction raster have been created. Those cells were then clipped to the river network, if existent, which reduces the number of points considerably. Once all potential pour points were calculated, a rating score (R) was assigned based on the following formula:

$$R = RA + 2RD \qquad (1)$$

where RA is the area difference and RD the distance ranking. RA was calculated by a relative comparison between the calculated watershed area and the original GRDC catchment area provided by the NMHS, with a value of 0 indicating a perfect match. A positive or negative deviation until 50 % was tolerated and assigned with the value 50, while all stations that exceeded this tolerance limit were initially sorted out and inspected later in detail manually. RD was derived from the distance of the station's coordinates to the respective pour point. Due to the double weighted distance ranking, stations are less likely to be chosen if they are far away. Since the buffer was 5 km large, a maximum value of 100 could be obtained. The two best results,

namely those with the best overall ranking calculated from MERIT and HydroSHEDS were chosen, and the catchment polygon

was calculated based on the one with the lowest R-value (Eq. 1). These stations were assigned as type "Automatic", while all other stations that exceeded the area difference of 50 were inspected manually and assigned as type "Manual". Manual procedures for station allocation are very time consuming as it involves the verification of river and station names as well as coordinates for each single station. The final decision on whether a station was moved to a new and "reliable" location was also based on the agreement of the calculated and reported GRDC catchment area.


**Table 2: Classification for the differentiation of quality levels of calculated catchments.**

| Type | Quality | Comment | Number of stations |
|------|---------|---------|--------------------|
| Automatic | High | Area difference <= 5 % and distance <= 5 km | 3918 |
| | Medium | Area difference 5-10 % and distance <= 5 km | 613 |
| | Low | Area difference 10-50 % and distance <= 5 km | 691 |
| Manual | High | Station and river name could be identified, area difference <= 5 % | 55 |
| | Medium | Station or river name could be identified, area difference between 5-10 % | 9 |
| | | Location seems correct, but GRDC area seems wrong | 18 |
| | Low | Stations were relocated manually, area difference between 10-50 % | 46 |
| | | Location seems correct, but catchment not well represented | 6 |

### 2.2.3 Catchment quality indicator

Even if a minimum quality is ensured with the help of the ranking assessment, the question for the user is how reliable the

derived station catchment areas are and how good the agreement is, since not all station catchment areas have the same quality. As the quality of the catchment boundaries depend on the accuracy of the provided coordinates as well as the catchment area, a further quality indicator has been assigned. The differentiation was based on whether the stations were generated automatically or manually and was further divided into three quality levels: "High", "Medium" and "Low". (Table 2). The quality indicator of catchments, which were calculated automatically, was assigned according to the calculated area difference

and distance. "High" quality was assigned for catchments with an area difference <= 5 % and a distance <= 5 km, "Medium" quality for an area difference 5-10 % and a distance <= 5 km, and "Low" quality for an area difference between 10-50 % and a distance <= 5 km.

The quality of manually calculated catchments was more complicated. Manually calculated catchments were assigned with a "High" quality index, if both the station and river name could be identified and the area difference is less or equal 5 %. If either

the station or river name could be identified but the area difference is between 5-10 %, catchments were classified into the "Medium" quality category. The "Medium" category was also assigned for cases, where the location seems correct, but it is

evident that the provided area is incorrect, since regional case studies or comparisons with upstream or downstream stations reveal discrepancies. "Low" quality involved catchments, where some of the stations were relocated manually and the catchment agreement tends to be mediocre between 10-50 %, but correlates to some extent with the flow behaviour. The

derivation was not always clear, especially for stations located in arid regions. In addition, catchments were assigned to the "Low" quality category, whose coordinates are correct, but the elevation model could not represent the catchment accurately. In total the catchment area for 5,356 stations was calculated, with 5,222 stations being allocated automatically and 134 manually inspected stations (Table 2).

## 2.3 Meteorological forcing data

The GRDC-Caravan extension contains the same 38 ERA5-Land time series features derived with the same algorithm as in the core Caravan dataset, see Table 1 and Table 2 in Kratzert et al. (2023). Note that the potential evaporation band from ERA5-Land, which is included in the core Caravan dataset and in this extension, is known to have issues, see e.g. Clerc-Schwarzenbach et al. (2024). For that reason, FAO Penman-Monteith Potential Evapotranspiration (PET) was recently added to the Caravan dataset, starting from version 1.5, see Kratzert et al. (2025). The PET time series are derived from ERA5-Land

time series using the code published by Singer et al. (2021). For consistency, however, the ERA5-Land potential evaporation band is still included in our extension, but it should be used with care.

## 2.4 Catchment attributes

The GRDC-Caravan extension includes the same 196 catchment attributes derived from HydroATLAS (see Table 3 and Table 4 in Kratzert et al. (2023), the same 10 climate indices derived from the meteorological time series (see Table 5 in Kratzert et

al. (2023), as well as the same metadata information (gauge latitude, longitude, area, country, and station name; see Table 6 in Kratzert et al. (2023)). Following the newly added FAO Penman-Monteith PET time series, all PET-related climate indices (4 in total) were additionally computed using the Penman-Monteith PET time series, resulting in a total of 14 climate indices. For better comparability, the climate indices were calculated from the same time period (1981-2022) as in the core Caravan dataset, even though longer records of forcing data are available.

Since the GRDC-Caravan extension also includes catchments that are much larger than the upper threshold that was used in the core Caravan dataset, which was 2000 $km^2$, we had to adapt the code that is responsible for deriving the catchment attributes from HydroATLAS on Google Earth Engine. The updated code has been merged into the official Caravan GitHub repository (Kratzert, 2025).

Additionally, the GRDC-Caravan extension includes a set of additional attributes that are specific to the GRDC data (see Table

1). Most of the attributes are derived from the GRDC station catalogue, available at the GRDC data portal (GRDC, 2024b). However, we made sure that the attributes related to streamflow availability are clipped to the periods included in this extension and that long-term streamflow indices are derived equally from the data that is included in the extension. Another attribute that

was added is nat_id, which is the national station code for each gauge station. This allows the user to find duplicates between GRDC station codes and stations that are already included in Caravan. The reason is that we also provide a mapping from the GRDC gauge ID to the national station IDs, which helps to filter out duplicates between GRDC stations and stations from other datasets already included in Caravan.

## 3 Dataset description

The GRDC-Caravan extension covers stations from 5356 catchments and 25 countries from the Global Runoff Database, spanning a time series length of 1 - 73 years (Fig. 2, Table 3). Most catchments are located in North America with 1133 stations in Canada and 999 stations in the United States. Followed by Australia (701 stations), Brazil (472) and South Africa (434). In addition, there is a large contribution by European countries such as Germany (336 stations), Norway (206), Spain (145), Austria (134), Finland (133) and Sweden (130). From other countries in Africa and South America only few stations could be collected. From Asia, no data could be provided at all.

Figure 3 shows the distribution of catchments and fraction of global landmass across the Global Environmental Stratification (GEnS) climate zones (letters "A-R") (Metzger et al. 2013). While most of the global landmass is lying in the 'extremely cold and mesic' ("F", about 16 %) or 'extremely hot and moist' area ("R", about 13 %), comparatively few catchments could be provided with our dataset in these regions (about 600 and 400 stations, respectively). In the 'cold and mesic' area ("G"), however, which accounts for the second largest fraction of global landmass (about 14 %), almost most of the catchments in the GRDC-Caravan extension could be provided (about 1400 stations). Second most stations are located in the 'warm temperate and mesic' area ("K", about 800). A large number of catchments in the dataset is also located in the 'cool temperate and dry' area ("H", about 600 stations) and the 'cool temperate and moist' ("J", about 400 stations), which make up a comparatively small area on Earth (4-6 %).

The size of the catchment areas ranges from very small streams with an area of about 2 km² up to the largest catchment area of the Amazon, covering 4,671,493 km² (Fig. 4). The majority of the stations (about 36 %) have a catchment area size of 1,000-10,000 km² followed by stations with catchment area sizes of 100-1,000 km² (about 30 %) and 10,000-100,000 km² (about 19 %). About 9 % of catchments are smaller than 100 km² and 5 % are larger than 100,000 km². The median size of the catchments is 2,015.8 km².

Figure 5 and 6 provide an overview about the distribution of streamflow records through time and their average lengths. Table 3 gives an overview about the average times series length by country and the data completeness. The highest number of gauging stations with streamflow data is available between 1980-2010 and more than 1000 stations provide a streamflow record of more than 70 years. The longest average time series length was provided by country data from the Czech Republic (68 years), Germany (64 years) and the United States (63 years). Shortest average time series were collected from Liberia (8.5 years) and Luxembourg (18 years). The average length of streamflow records is about 50 years. In total, the extension covers a total of 275,444 years of records.

The dataset shows a high data completeness of more than 90 % with only few missing values. Lowest completeness was observed in Jamaica (82 %) and Canada (85 %). Missing values are also responsible for annual cycles in data availability in Figure 5, which originate from snow- and ice-dominated regions during winter. In the same figure, the strong decline of streamflow records since 2020, however, must not be misinterpreted. It is not an indication of a decreasing monitoring

infrastructure, but is due to the fact that GRDC only includes quality controlled (yearbook) river discharge data from NMHS. Since many countries take several years for the final release of the quality-controlled data, this time lag until GRDC can release quality-controlled data with confidence is evident in the drop-off of the number of available stations released by NMHS.

In comparison to the core Caravan dataset of Kratzert et al. (2023), the GRDC-Caravan extension strongly increases the global coverage of the dataset and takes the total number of Caravan catchments to 22,372. While in the core Caravan dataset mostly

catchments from Northern America, Central Europe and South America were included, the new extension significantly improves the spatial coverage in Northern and Southern Europe, South America, South Africa, Australia and New Zealand. However, there are still many data scarce regions across Africa and Asia (Fig. 2).

The comparison across climate zones between the core dataset and the extension shows that the majority of catchments in Caravan is now available from the 'cold and mesic' area (letter "G", Fig. 3). The second largest portion is contributed by the

'warm temperate and mesic' area ("K"), followed by the 'cool temperate and moist' ("J") and 'cool temperate and dry' area ("H"). Although the extension contributes a distinct number of stations in the 'extremely cold and mesic' ("F") and 'extremely hot and moist' area ("R"), these regions are still underrepresented despite their total fraction of global landmass.

With respect to temporal coverage, the GRDC-Caravan extension increases the time series of Caravan from 40 to 70 years (660,382 years). The inclusion of pre-1981 data into the GRDC-Caravan extension, however, is now possible as Google Earth

Engine now includes the full record of ERA5-Land data. With respect to the catchment size, in the core Caravan data set only gauges with a total drainage area between 100 km² and 2000 km² have been included (Kratzert et al. 2023). In the GRDC-Caravan extension also smaller and much larger catchments are included. This together with the extended time series is particularly important for studies investigating climate effects on global river basins.

Although the GRDC-Caravan extension supplied several new stations to the dataset, there was also an overlap of 1589

duplicates between the core and the extended datasets. This includes 202 stations from Brazil (CAMELS-BR), 944 in North America (HYSETS), 69 from the US (CAMELS (US)), 191 from Australia (CAMELS-AUS), and 183 in Central Europe (LamaH-CE). There are a few gauges where a single GRDC gauge maps to both, a CAMELS (US) and a HYSETS gauge, which are counted as two duplicates in the value 1589. The high number of duplicates were still included in the expansion because they add an average of 13.8 time series years.


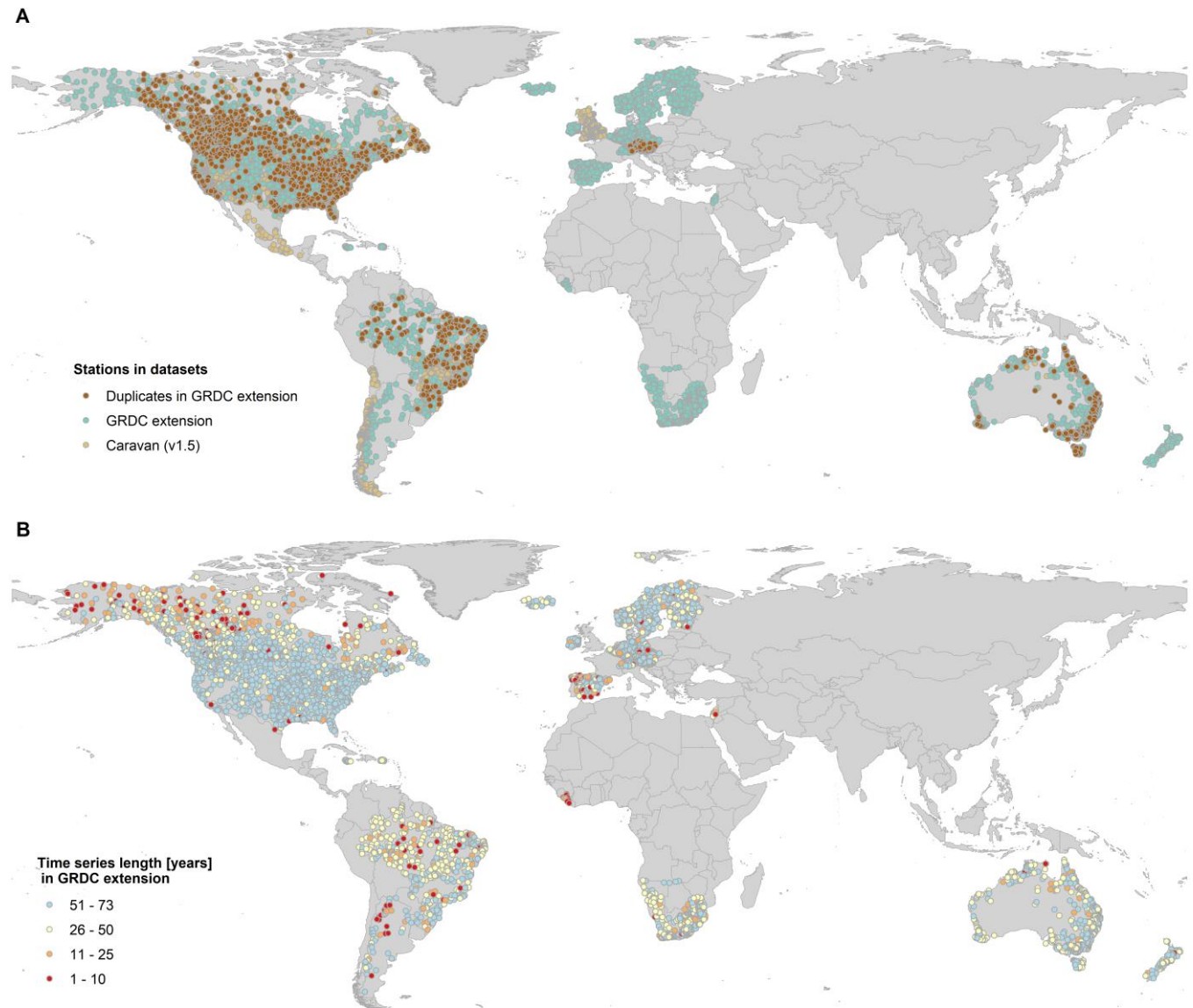

**Figure 2: A) Global distribution of stations included in the core Caravan dataset (beige, Kratzert et al. 2025) and in the GRDC-Caravan extension (green, this paper). Brown shows duplicate stations between the core dataset and the extension. B) Time series length of stations included in the GRDC-Caravan extension. Note that the data record in the Caravan extension starts 1951 (because of the forcing data). Longer records for individual stations might exist on the GRDC data portal (GRDC, 2025c).**



**Table 3: Overview of country data included into the dataset. From WMO Region II (Asia) no data is provided in the extension.**

| Region/Country | Number of stations | Average times series length [yrs] | Data completeness [%] |
|---|---|---|---|
| **WMO Region I: Africa** | | | |
| Liberia | 9 | 8.5 | 98.3 |
| Namibia | 51 | 45.2 | 92.8 |
| South Africa | 434 | 48.4 | 93.4 |
| **WMO Region III: South America** | | | |
| Argentina | 59 | 43.0 | 94.7 |
| Brazil | 472 | 41.7 | 93.1 |
| **WMO Region IV: North America, Central America, Caribbean** | | | |
| Canada | 1133 | 44.7 | 85.4 |
| Jamaica | 12 | 51.9 | 82.2 |
| Puerto Rico | 24 | 45.4 | 95.5 |
| United States | 999 | 63.3 | 95.6 |
| **WMO Region V: South-West Pacific** | | | |
| Australia | 701 | 49.8 | 97.0 |
| New Zealand | 66 | 50.4 | 98.4 |
| **WMO Region VI: Europe** | | | |
| Austria | 134 | 54.6 | 99.7 |
| Belgium | 55 | 34.0 | 96.8 |
| Czech Republic | 29 | 68.1 | 99.6 |
| Estonia | 51 | 49.5 | 89.4 |
| Finland | 133 | 54.3 | 99.1 |
| Germany | 336 | 64.1 | 99.8 |
| Iceland | 23 | 55 | 95.9 |
| Ireland | 43 | 54.6 | 92.9 |
| Israel | 8 | 33.3 | 97.2 |

| Luxembourg | 11 | 18.0 | 100 |
| Norway | 206 | 51.8 | 96.4 |
| Spain | 145 | 41.8 | 91.5 |
| Sweden | 130 | 57.8 | 98.6 |
| Switzerland | 92 | 56.2 | 99.8 |

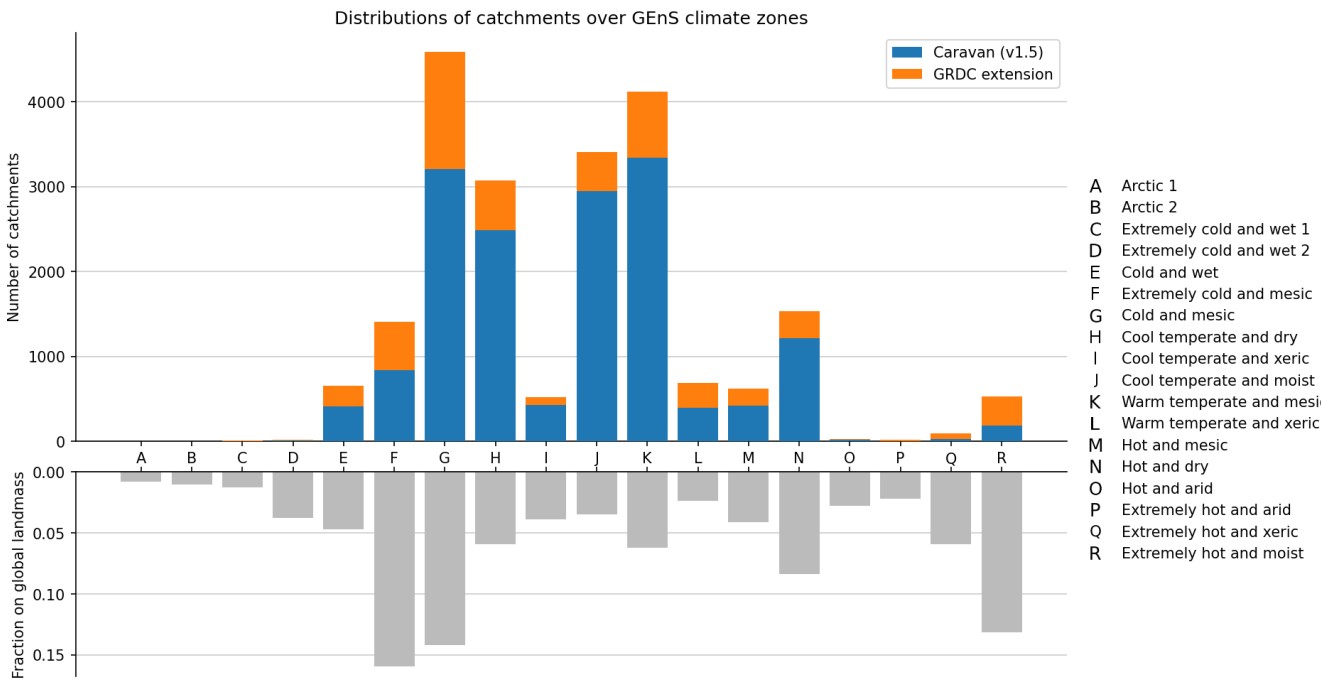

**Figure 3: Distribution of catchments among the Global Environmental Stratification (GEnS) climate zones (Metzger et al., 2013). Blue bars denote catchments included in the Caravan core dataset (v1.5, Kratzert et al. 2025) and orange bars denote catchments included in this dataset. The bottom part of the plots shows the fraction of a particular climate zone on the total land mass.**

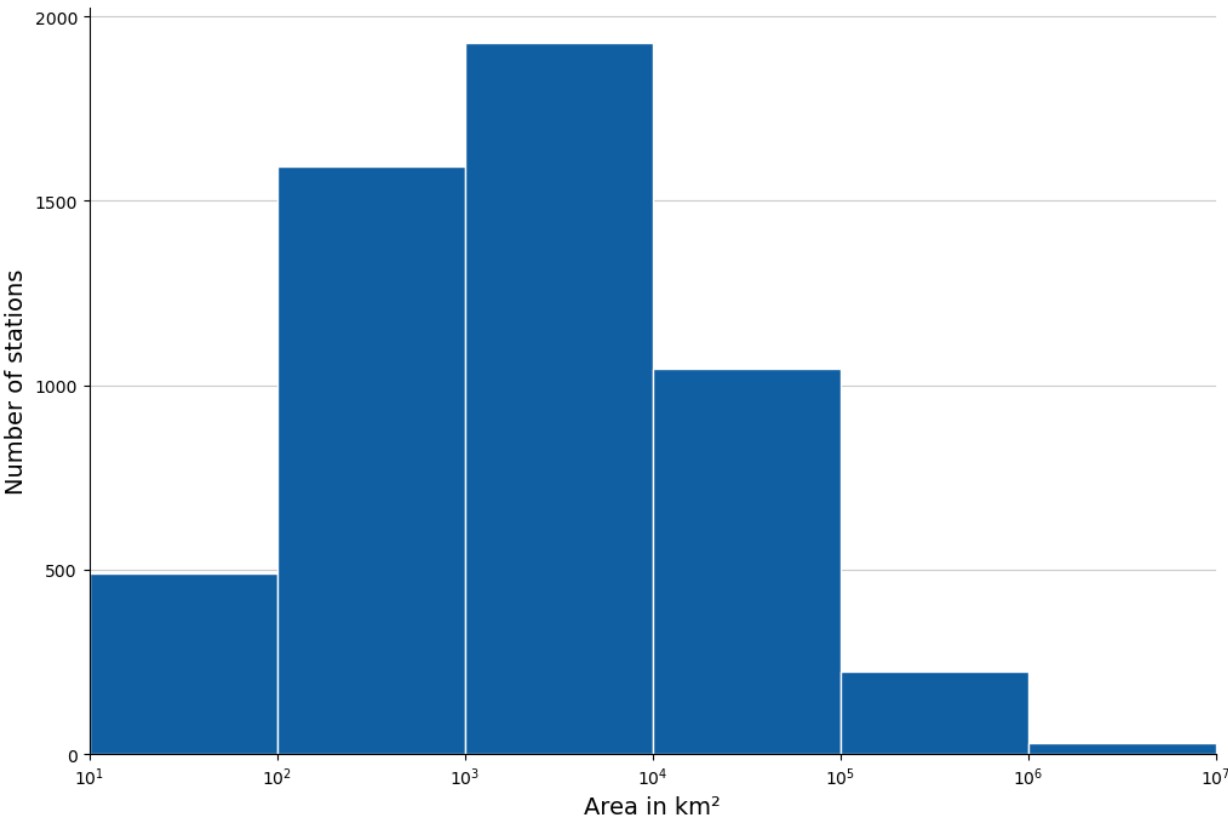


**Figure 4: Histogram showing the number of stations and their corresponding catchment area.**

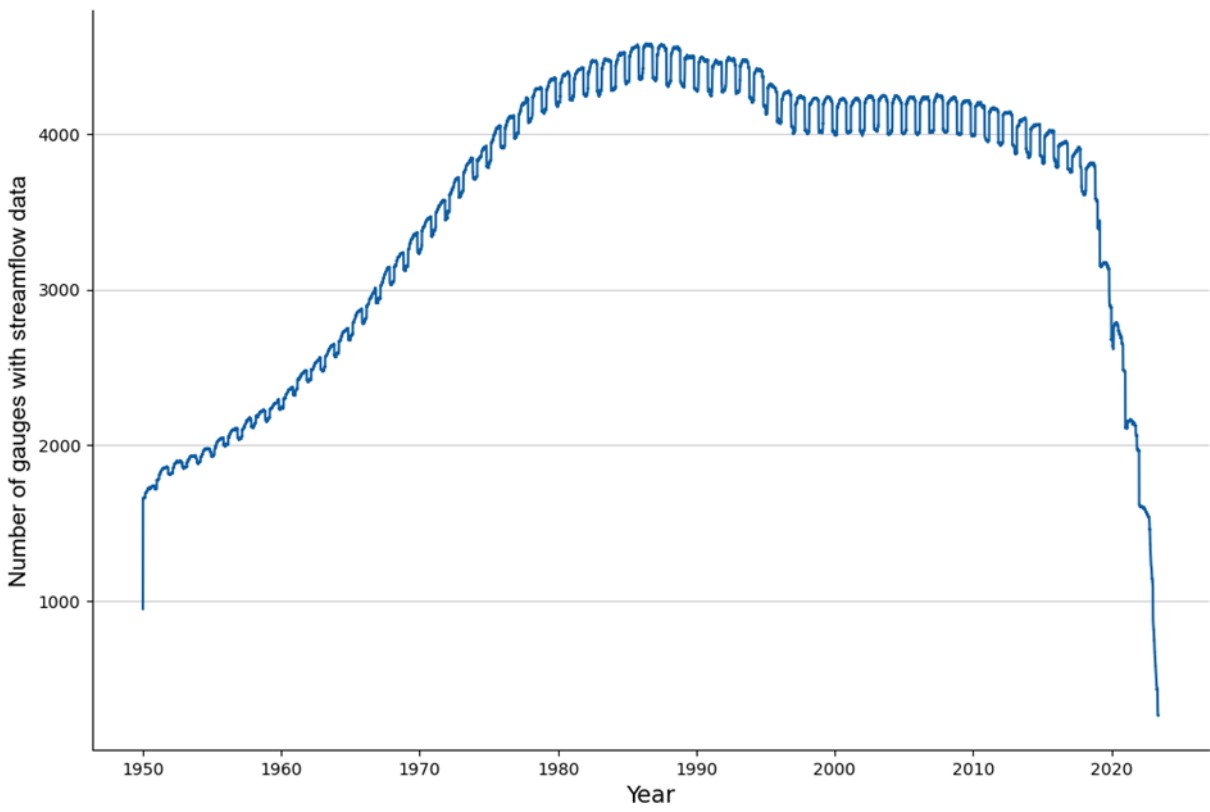

**Figure 5: Number of gauging stations with streamflow data through time. The recent drop-off is not an indication of a decreasing monitoring infrastructure, but rather is evidence of the time it takes until GRDC receives data from NMHS and can release quality-controlled data. The annual cycles in the data availability are due to missing data in snow-dominated regions during winter.**


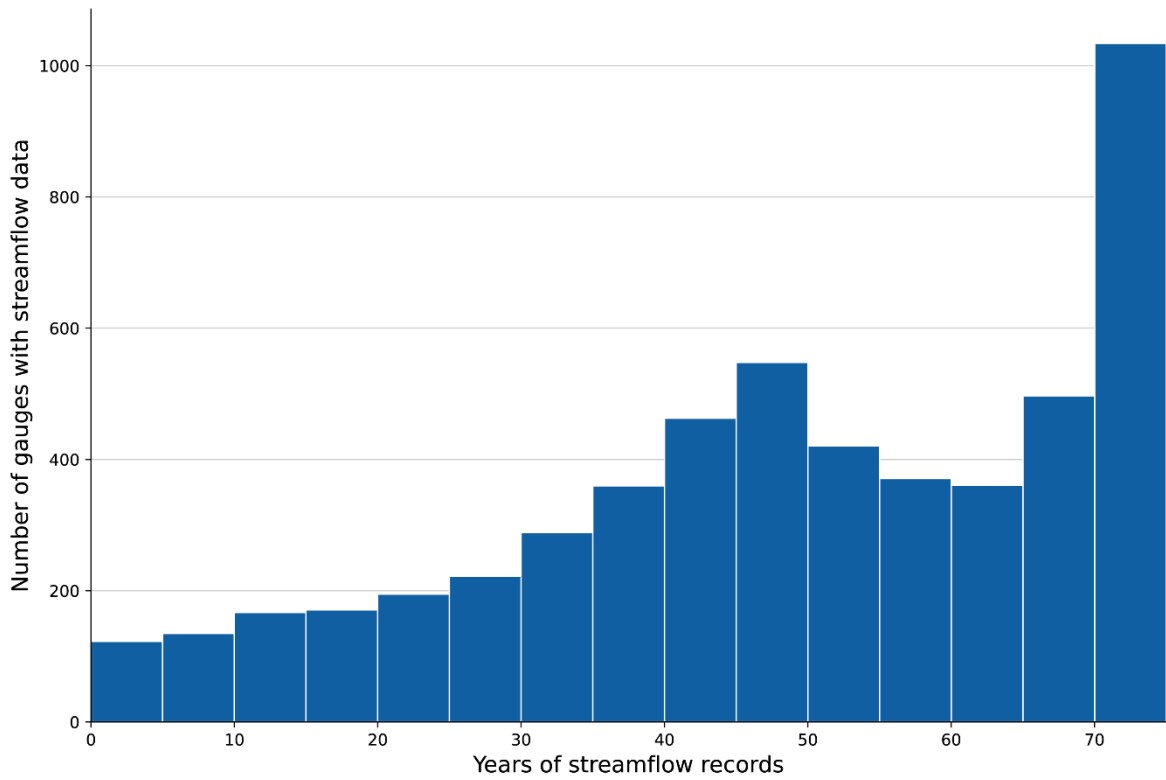

**Figure 6: Number of gauging stations by years of with streamflow record over time. The average times series length is of about 50 years.**

## 4 Data and code availability

The GRDC-Caravan extension dataset is publicly available on Zenodo: https://doi.org/10.5281/zenodo.15349031 (Färber et al., 2025). The original code to produce Caravan extensions is available here: https://github.com/kratzert/Caravan (Kratzert 2025). The R package 'GRDCFlowTools' can be accessed here: https://github.com/bafg-bund/GRDCFlowTools (GRDC, 2025b).

## 5 Outlook

This paper introduces a contribution to the Caravan initiative extending the core data set in space, but also in time. Although some countries are already covered in the core Caravan dataset, GRDC stations for the same countries such as the USA, Canada, Brazil, and others have been added because the GRDC time series have brought an additional 70 years of discharge data for over 700 stations. On average for duplicated basins we add 13.8 years. We show that by using the Caravan platform, flow data from any set of catchments can be augmented using hydrometeorological time series and catchment attributes and

formatted in a way that enables its immediate use as part of the Caravan dataset. This benefits the community and because the extraction process is automated, additional catchments can easily be added.

We feel it is useful to step back and look at the progress the field of LSH has made to appreciate the importance of this extension and of the Caravan project more generally. A decade ago, open datasets providing hydrometeorological time series and attributes describing the landscape and human influences in large samples of catchments barely existed (Gupta et al. 2014). This impeded the systematic analysis of changes across hydroclimatic gradients, levels of human interventions and decades. Since then, significant community efforts have been underway to collate such datasets and make them publicly available. It resulted in data from multiple countries entering the public domain, adhering to a common blueprint formed by early datasets (MOPEX and CAMELS). Caravan and its extensions (including this one) currently encompass spatially consistent time series and attributes from 22,494 catchments from 35 countries, covering a total of 660,382 years of records, all published under an open license, free for the community to use. LSH datasets have enabled hundreds of hydrological studies across the globe, their use keeps increasing and we believe they will keep playing a key role in hydrological sciences for the foreseeable future. Without the 25 WMO member states who provided their data under an open access license, the publication of the GRDC Caravan extension would not have been possible - most of the data in the Global Runoff Database is still published under a data policy which requires identified access, prohibiting redistribution and commercial use. As GRDC is continuously receiving new data from NMHS, it is intended that this dataset will be updated. We hope that in the future more and more member states will make their data available under open licenses so that it can be included in future versions of the extension. Future versions of the extension will also benefit from upgrades of the Caravan dataset, such as the recent inclusion of weather forecasts/hindcasts (Shalev & Kratzert 2024a, b). Overall, we see these future developments as two-way exchanges between this extension and the wider Caravan dataset: this extension extends Caravan and, at the same time, will benefit from future Caravan improvements. Extending the dataset in space and time are just two ways to contribute to Caravan, and here we invite members of the community to imagine and share other extensions, for instance including time series for new variables or new landscape attributes.

**Author contributions**

HP, FK, NA, GS, SM and UL were responsible for the design and processing of the dataset; CF, HP, FK and NA organized and wrote the manuscript. All authors discussed the results and contributed to the final paper.

## Competing interests

The authors declare that they have no conflict of interest.

## Acknowledgements

GRDC thanks the World Meteorological Organization (WMO) for providing support and the Federal Institute of Hydrology, Germany (BfG) for funding. Thomas Recknagel is acknowledged for providing the R package 'GRDCFlowTools'. The NMHS are acknowledged for providing their valuable data sets.

## Review statement

This paper was edited by Sibylle Hassler and reviewed by Thiago Nascimento and one anonymous referee.

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
