# Peer review of "GRDC-Caravan: extending Caravan with data from the Global Runoff Data Centre"

_Earth System Science Data, 2024_

## Referee Comment (RC2)

385

[referee-annotated manuscript omitted]

---

## Author Comment (AC1)

**Response to Reviews**

*We thank the reviewers for their thorough and constructive evaluation and feedback. We have tried to respond to all comments as well as possible. Please find our reply to RC1 below.*

**RC1**

This manuscript introduces the GRDC-Caravan dataset, which extends the Caravan hydrology dataset by incorporating discharge data from the Global Runoff Data Centre (GRDC). By expanding spatial and temporal coverage to over 5,000 stations in 25 countries, it's clear that this work makes an important contribution to large-sample hydrology (LSH). The dataset is particularly valuable for underrepresented regions, providing a solid foundation for global hydrological research while aligning with open science principles. Personally I feel impresed by both the valuable job of GRDC as institution and the intitiative of the team to further contribute with this Caravan extension. I am sure thart this will provide a powerfull new dataset for the hydrology community. While the manuscript is undeniably significant and deserves publication, there are a few minor areas that need improvement to meet the standards of *Earth System Science Data* (ESSD). Addressing these issues will help improve clarity, structure, and reproducibility. With these adjustments I believe that the paper will be ready for the journal.

*Thank you very much for your positive feedback. We have revised the manuscript according to your comments in order to improve the clarity, structure and reproducibility of the paper.*

Suggestions for improvement

**1. Manuscript Structure:**

The manuscript lacks a clear distinction between methods and results, making it difficult to follow.

Please consider splitting the "Dataset Description" section into two separate sections: "Methodology" and "Dataset Description."

*Thank you for this helpful suggestion. The point has also been addressed in the second review and we followed this suggestion. The manuscript now contains a chapter "Methodology" and "Dataset Description". We left most of the previous chapter in the Methodology and shifted only the parts, which were describing the dataset itself to the new chapter. The Dataset Description was improved and contains now a better description of the dataset itself as well as a comparison to the core Caravan dataset.*

**2. Figures and Captions**

I have the impression that while the figures are visually informative, they lack context and detailed descriptions.

Improve captions to explain the figures' relevance to the dataset's purpose

*We have followed this suggestion as best as possible. We have critically reviewed the figure captions and have tried to improve them where necessary.*

Provide more discussion of the figures in the main text to guide readers

*We have followed this suggestion and have improved the description and linkage of figures in the text.*

**3. Comparisons with Existing Datasets**

I feel that the paper presents a limited discussion of how GRDC-Caravan compares with other LSH datasets.

Add a detailed overview of LSH releases. Where did it start? MOPEX? CAMELS? Which are the main datasets currently available? Which countries are covered? How did Caravan arrive to existence? What about other datasets and similar initiatives? GSIM? How is the situation of LSH datasets by the end of 2024?

*We have critically examined this point and would like to point out that a brief review of the mentioned history of LSH and main datasets (including MOPEX, GSIM, CAMELS and Caravan extensions) has been already provided in paragraph 2 and 3 of the introduction. We have also added text to the outlook section (see second paragraph) to summarise the current state of LSH and the importance of LSH datasets. We think that a full review of LSH would go beyond the scope of this data description paper. However, we improved the existing review and tried to highlight the link of the GRDC-Caravan extension with existing initiatives.*

Include a comparative analysis highlighting GRDC-Caravan's unique contributions (e.g., coverage in underrepresented regions, temporal/spatial resolution, data quality). Remember that you should sell the idea of why GRDC-Caravan matters.

*We understand that the uniqueness of the GRDC-Caravan extension did not become clear in the initial submission. We have extended the abstract and the introduction with a better explanation how GRDC-Caravan significantly contributes to the spatial coverage of Caravan, including the extension of the times series to 70 years. A more detailed description and comparison to the core Caravan dataset is now also provided in the updated chapter "Dataset description". We have also updated Fig. 2 and 3, which include now also a comparison between our dataset and the core Caravan dataset.*

**Data Records and Repository Description**

I feel that there is still insufficient details about the organization of the Zenodo repository and dataset files.

Describe the repository structure, including folders and file organization. Of course, a further description of the data included, and units is already provided by Caravan, but for users, it is always beneficial to know what to expect after the download.

*We refer the comment to the description of the repository in the manuscript and not the organization of the repository itself. The organization of the repository has been already improved with a data description file before the review (requested by the editor). However, we followed the comment and added a subchapter on "Dataset structure" in the "Methodology" in order to make the organization of the repository and dataset files clearer. Please note that version 0.4 of the Zenodo repository now also includes a preview of the files (https://zenodo.org/records/14725028).*

L245: Specify the version number of the dataset being referred to on Zenodo. Pay attention to always place the specific version being described in the paper, not the general DOI for all versions.

*We have checked the doi in the manuscript. It is referring to the specific version 0.4 described in the paper (10.5281/zenodo.14725028), not the general doi (10.5281/zenodo.8425586).*

**5. Reproducibility and Code Availability**

The authors state that the GRDCFlowTools R package is only available upon request, limiting reproducibility.

If possible, publish the package on a public repository (e.g., GitHub or CRAN).
Include usage examples to facilitate adoption and reproducibility.

*We have now published the package on GitHub in order to increase the reproducibility. It also includes usage examples (https://github.com/bafg-bund/GRDCFlowTools).*

**6. Duplicates**

While the authors acknowledge, the duplicated catchments are not explicitly labeled in the dataset.

If possible, consider  adding a column that identifies duplicated catchments in the dataset.

*We have discussed this issue and have decided not to place the duplicates in a separate column in the dataset. However, the duplicates are now explicitly labelled in Fig. 2 and are better explained in the dataset description. The new subchapter "dataset structure" also explains how duplicates can be found based on the file national_station_ids.csv:*

*"The attribute 'nat_id', which is the national station code for each gauge station, allows the user to find duplicates between GRDC station codes and stations that are already included in Caravan"*

**7. Dataset Analysis and Validation**

This paper already covers a clear description of the dataset being published. But to be fair to other similar dataset papers published in ESSD, it indeed lacks a bit of a overview and preliminary analysis of such dataset, which could also act as a technical validation of what is being published. While I see that the GRDC makes available data which has been checked by the NMHS, I think that future users could benefit from a general overview, which could also act as a validation. Personally,

I do not see the need to perform any modelling analysis with the data as previous artiodactyls papers have done.

Include plots showing spatial distributions of key streamflow signatures and climatic indices (e.g., Q mean, BFI, aridity).
Try to include a creative plot showing an overview of this new discharge data for users. Why not some small boxplots of the Q mean covered among the GEnS climatic zones? Something inspired in Figure 4 by Marvin Höge et al. 2024 in CAMELS_CH? Take this as a suggestion (inspiration), not a requirement.

*We have thoroughly discussed this comment and have decided not to conduct further dataset analysis and validation. We think that an in depth analysis of the dataset would go beyond the scope of a data description paper as defined by ESSD:*

*"Although examples of data outcomes may prove necessary to demonstrate data quality, extensive interpretations of data (...) remain outside the scope of this data journal. ESSD data descriptions should instead highlight and emphasize the quality, usability, and accessibility of the dataset (...) and should describe extensive carefully prepared metadata and file structures at the data repository."*

*We think that Fig. 3 already gives a similar overview as Höge et al. 2024, showing the distribution of our dataset along climate zones. While Höge et al. 2024 are providing a regional dataset (CAMELS-CH), our dataset is on the global level and we consider a global map of catchments too difficult to interpret. However, we have improved the description of the dataset in the text in order to better explain its composition and spatio-temporal contribution.*

Validate catchment boundaries with visual aids or references to prior successful applications of MERIT Hydro and delineator.py.

*We have followed this comment and provided prior successful applications of MERIT Hydro and delineator.py in the text (l. 154).*

**8. Outlook Section**

The outlook could be more concise and impactful.

Consider Emphasizing more the televance of GRDC-Caravan for global hydrology. Describe the need for new Caravan extensions, and the Importance of open data policies from NMHS. Streamline the section to improve rhythm and flow.

*We followed the comment and have improved the outlook section.*

Specific points:

General:

Some sentences are long and complex, making them hard to follow. Please while reviewing the manuscript make sure to go through such points. For example:

L 171: Consider rephrasing "After the calculation of all potential pour points was completed, a rating score R was calculated..." to: "Once all potential pour points were calculated, a rating score (R) was assigned based on the following formula."

*The sentence has been rephrased as suggested.*

Introduction

L33–34: Please consider to rephrase "is the primary objective of the principal reason" to something in the lines of "The primary objective of the Global Runoff Data Centre (GRDC) is to (…)."

*The introduction has been rephrased and the sentence replaced.*

Dataset description

L115: Acknowledge that Figure 3 adapts elements from the original Caravan publication. Add a note in the caption, e.g., "This figure is adapted from the original Caravan publication (Kratzert et al., 2023)."
Citation: https://doi.org/10.5194/essd-2024-427-RC1

*We have improved the figure and have added information from the core Caravan dataset with a reference to the original publication. Therefore we think that a further note on figure adaption is not necessary.*

---

## Author Comment (AC2)

**Response to Reviews**

*We thank the reviewers for their thorough and constructive evaluation and feedback. We have tried to respond to all comments as well as possible. Please find our reply to RC2 below.*

**RC2**

This is my first review of the manuscript by Färber et al. The manuscript addresses a relevant topic and aligns well with the journal's scope. Utilizing a river discharge dataset with an extended record length is valuable for numerous research applications. However, I found the paper to be poorly organized at times and not always clear in its presentation. Furthermore, it lacks a thorough discussion of the novel contributions of this dataset, remaining overly generic in certain aspects.

*Thank you very much for your constructive feedback. We have revised the manuscript according to your comments in order to improve the structure and clarity of the paper. We have also tried to show better the uniqueness and novelty of our dataset.*

**Abstract**

The abstract is unclear and should explicitly state the manuscript's objective, which only becomes apparent after a careful reading of the paper.

*We have revised the abstract in order to make the objective of the paper clearer.*

**Introduction**

The introduction includes essential material but could benefit from improvement, particularly in its concluding section, which is overly cryptic. The introduction should conclude with a clear and well-articulated statement of the manuscript's objective.

*Similar to the abstract, we have revised the introduction in order to better explain the objective of the manuscript.*

**Dataset Description and Methods**

The manuscript conflates the dataset description with the methods. These should be clearly separated for better clarity. For instance:

Figure 1 is not cited in the text.

*Please check, Figure 1 is cited in the introduction (l. 58) and the methodology (l.72).*

River discharge data should separately describe the current GRDC dataset and the Caravan dataset.

*We followed the comment and added separate chapters on methodology and dataset description. The comparison between the core Caravan dataset and the GRDC extension is now improved and also highlighted in Fig. 2 and 3.*

Catchment attributes and DEM should be included as part of the dataset description.

*We think the detailed description of the methodology about the catchment attributes is sufficient and that an analysis about it would go beyond the scope of this data description paper.*
*Since the DEM is an input data for the area calculation, the short overview of the extent and the resolution seems adequate.*

**Outlook Section**

The outlook section lacks a crucial comparison between the datasets prior to and after the extension. It should emphasize what has been improved, the capabilities now available that were previously unattainable, and the added value of this new dataset.

*We have followed the comment and have improved the outlook section.*

**Additional minor comments are reported in the annotated pdf:**

The abstract is not really informative of what is the aim of this paper and what is finally provided by this new dataset or this new extension. Please try to improve it.

*We have revised the abstract in order to make the aim and objective of the paper more clear.*

a new extension of what?

*The sentence has been revised. A better explanation of "extension" is now provided in paragraph 3 of the introduction.*

this sentence is not clear.

*This sentence has also been revised.*

Please explain how

*This sentence has been removed. A detailed description of how river discharge data and catchments are compiled is provided in the methodology.*

This is not very detailed. Maybe refer to the section where this is explained

*We have improved the sentence.*

This is more method than dataset description

*We have followed the reviewers comment and reorganized the manuscript into "Methodology" and "Dataset description". The figure is now part of the chapter "Methodology".*

Please clarify "base" dataset

*In the revision, we have agreed to use "core" dataset instead of "base". The definition of "core dataset" is now explained in paragraph 3 of the introduction.*

Can you explain how you handle catchments with areas lower than the grid size of era land?

*We take the forcings of the grid cell (i.e. the spatial averaged forcings of the catchment area are just defined by the forcings of that grid cell).*
*There is some degree of uncertainty, because e.g. the "rain" of a grid cell could fall entirely on the area outside of the polygon. For this very reason, the initial versions of the Caravan core dataset had a min (and also max) area defined. However, this was removed in a later version of Caravan and we follow the same principle here. With this dataset we provide a dataset to people that naturally comes with uncertainties. People using this dataset and working in environmental sciences in general, should probably be aware of the uncertainty that comes with this and depending on their research study decide which of the basins to include.*

---

## Author Response (AR2)

**ESSD-2024-427 | Data description paper**

GRDC-Caravan: extending Caravan with data from the Global Runoff Data Centre

Claudia Färber, Henning Plessow, Simon Mischel, Frederik Kratzert, Nans Addor, Guy Shalev, and Ulrich Looser

**Reply to Second Review (Minor Revision)**

Dear Dr. Hassler,

Dear Reviewers,

Thank you very much for your valuable comments.

According to your feedback, we have made the following changes:

- Line numbers were added in the initial reply on RC2
- The data description file in the Zenodo archive has been updated, containing now all tables, table headers and units that are provided in the dataset

We hope that everything has been implemented to your convenience.

Kind regards

Claudia Färber and co-authors

---

## Author Response (AR3)

**ESSD-2024-427 | Data description paper**

GRDC-Caravan: extending Caravan with data from the Global Runoff Data Centre
Claudia Färber, Henning Plessow, Simon Mischel, Frederik Kratzert, Nans Addor, Guy Shalev, and Ulrich Looser

**Reply to Justification (Minor Revision)**

Dear Dr. Hassler,
Thank you very much for your valuable comments and questions. Please see our response below.

**Comments:**
1) Data description, Table 1 and the respective dataset: It takes quite a while to find the column names in the csv in the table due to the different orders and the names including numbers for different months, classes, etc. that are not sorted together. It would help if you had a) the same order in both the csv and the Table 1, b) put the logically related tables in columns together (e.g. _1 to _12 of the same attribute), or everything that is part of the "hydrology" part9, and c) if you put the attribute name also in a separate column in Table 1, similar to Tables 2-4. And it seems the column name for "Land surface runoff" is missing?
   We have resorted Table 1 and the respective dataset. Both are now sorted in alphabetical order, but we kept the classification into thematic groups in the data description file to provide the users a better overview what is available. The attribute name is now in a separate column. The column name for "Land surface runoff" has been corrected.
2) Data description tables in general: It's quicker to relate the tables in the description to the csv files, if you also put the file name in the table header. Also: for completeness you could add the gauge_id to all tables - at the moment it is a bit random that you have a separate paragraph on it that is meant for all attribute tables, but nevertheless it shows up in Table 4. Simply adding it to all tables as first column would simplify things.
   We have fully addressed this comment and have added the file path into the table header and the gauge_id as first row to all tables.
3) Data description Table 2: The names probably refer to an older version of the dataset. As you're describing the one including the Penman-Monteith estimates in the ESSD Paper and refer to that version, the column names in Table 2 need to be updated.
   We apologize for this issue. The table has been updated accordingly.
4) side note: The header of Table 3 could be more descriptive, as "metadata" pretty much refers to all information in the whole data description. Maybe "location attributes" or something like that would be more specific?
   The table header was changed to "gauging station attributes".

**Questions:**
1) In the license file there is one license from Belgium that says "no sharing with third parties allowed". Can this data still be shared publicly as part of the dataset?

We apologize for this issue. We have contacted the Hydrological Service of Wallonia again to clarify the distribution of the data in GRDC and the present dataset. They confirmed the permission to share the data. The license file has been updated accordingly.

2) In the netCDF time series: was there a specific reason why the files are not cf conform? (as sticking to the standard names etc. usually helps with using, understanding and plotting the netCDFs…)
You are right that following the conventions would have been better. However, the used NetCDF format has been defined in the original Caravan dataset and our dataset is an extension to it. Changing it now, would complicate the compatibility between the different Caravan extensions. Therefore, we followed the naming conventions and structure of the Caravan dataset.